# Multi-scale V-net architecture with deep feature CRF layers for brain extraction
Jong Sung Park [1] ✉, Shreyas Fadnavis[2] & Eleftherios Garyfallidis [1]

## Abstract

**Background** Brain extraction is a computational necessity for researchers using brain imaging data. However, the complex structure of the interfaces between the brain, meninges and human skull have not allowed a highly robust solution to emerge. While previous methods have used machine learning with structural and geometric priors in mind, with the development of Deep Learning (DL), there has been an increase in Neural Network based methods. Most proposed DL models focus on improving the training data despite the clear gap between groups in the amount and quality of accessible training data between.

**Methods** We propose an architecture we call Efficient V-net with Additional Conditional Random Field Layers (EVAC+). EVAC+ has 3 major characteristics: (1) a smart augmentation strategy that improves training efficiency, (2) a unique way of using a Conditional Random Fields Recurrent Layer that improves accuracy and (3) an additional loss function that fine-tunes the segmentation output. We compare our model to state-of-the-art non-DL and DL methods.

**Results** Results show that even with limited training resources, EVAC+ outperforms in most cases, achieving a high and stable Dice Coefficient and Jaccard Index along with a desirable lower Surface (Hausdorff) Distance. More importantly, our approach accurately segmented clinical and pediatric data, despite the fact that the training dataset only contains healthy adults.

**Conclusions** Ultimately, our model provides a reliable way of accurately reducing segmentation errors in complex multi-tissue interfacing areas of the brain. We expect our method, which is publicly available and open-source, to be beneficial to a wide range of researchers.

## Plain language summary

Computational processing of brain images can enable better understanding and diagnosis of diseases that affect the brain. Brain Extraction is a computational method that can be used to remove areas of the head that are not the brain from images of the head. We compared various different computational methods that are available and used them to develop a better method. The method we describe in the paper is more accurate at imaging the brain of both healthy individuals and those known to have diseases that affect the brain than the other methods we evaluated. Our method might enable better understanding and diagnosis of diseases that affect the brain in the future.

Brain MRI computes the tissue composition to create an image of the brain. However, other parts of the body also exist within the image. Unnecessary non-brain tissues could include the face, eyes, spine, etc. For additional methods such as registration, tractography and tissue segmentation to work properly, it is important to have the image of the sole brain without other parts of the body that will disrupt the algorithm or analysis. Accurate manual segmentation (gold standard) can be ideal, but it is very time-consuming with in need of an expert, who can also include biases in the mask. Thus, automatic brain extraction, also called skull stripping is a semantic segmentation task necessary to pre-process the brain to perform forthcoming analyses. Despite the simple explanation of the problem, difficulties arise from several factors of the MR images. Non-brain tissues can be spatially close to the brain and have very similar intensities, especially white matter in T1-weighted images. These issues are troublesome for current algorithms. Artifacts and noise common in T1 weighted images can also disrupt the robustness of the methods.

Despite the fact that multiple modalities can be used for brain extraction, in this paper we focus on T1 weighted brain MRI, as these are the most commonly used images for brain extraction and they are nearly always acquired with each scan.

[1]Intelligent Systems Engineering, Indiana University Bloomington, Bloomington, IN, USA. [2]Massachusetts General Hospital, Harvard Medical School, Boston, MA, USA. ✉e-mail: pjsjongsung@gmail.com

## Traditional methods

Traditional approaches in skull stripping use known priors of the brain. They can be categorized by what type of prior the algorithm relies on: morphology, intensity, surface structure, template matching or a combination of these[1]. While every method in any category has its ups and downs, we will discuss a few methods that are popularly used by experts and work on T1-weighted images. Rehman et al.[1] provides a good summary of the various methods proposed in the literature.

Brain Extraction Tool (BET)[2], one of the most used methods, uses the intensity and brain surface structure as its prior. It creates an initial sphere using the intensity differences per tissue type. The sphere is deformed in each direction to form the brain's shape. Even though the model is fast and does not need any preprocessing beforehand, it has been known to have limitations in the segmentation results[3]. 3dSkull-Strip used in Analysis of Functional NeuroImages (AFNI)[4] improves BET by adding a few modifications that help exclude the eyes and the ventricles in the image, removing some of the false positive tissues. Freesurfer's method[5] adds an intensity/structure prior, using the fact that white matter has a higher intensity in the T1 modality and is one connected segment, to create an initial volume and later deforming it to match the brain surface. ROBEX (robust, learning-based brain extraction system)[6] uses a more hybrid approach of creating an initial mask by using a random forest classifier on extracted features, which is fit/deformed to match a more accurate surface.

Non-deformable suface-based models are widely used as well. Brain Surface Extractor (BSE) uses anisotropic diffusion filters to smooth out low-contrast edges followed by edge detection with morphological erosion. The edges are later expanded to match the brain. BEaST[7] uses patchwise comparison to a set of templates pre-constructed from 80 segmented and manually corrected brain masks as a prior. It also uses multi-resolutions of patches to optimize the computation and reduce false negatives. ALFA (accurate learning with few atlases)[8] registers manually segmented neonatal brain atlases to the target image, selects the few closest to the average of them and fuses the labels using machine learning approaches to create the segmentation output.

Since the use of Alexnet[9] for image classification tasks, there have been many developments in using Deep Learning models for images[10–12]. However, there are several reasons why it is often not optimal to implement the model architecture and ideas directly to medical images. First is the lack of data. While it is relatively easy to acquire computer vision data that can come from various input devices, cameras, satellite images, videos etc, that is not the case for medical images. Since these contain personal information, it is difficult to gather the data in the first place and even harder to make the data public due to patient protection regulations and governmental guidelines. Hence, a lot of them cannot be used or shared among other research teams to train and evaluate their models. Additionally, labeling the data demands careful analysis of the images by experts, thus there are fewer public data with true ground truth labels. Most ground truth labels provided in the public datasets are mainly created by another machine learning method with manual editing[13–16]. The dimension of the data is another obstacle since most of the data are 3D, and even a single T1-weighted modality image is a few hundred computer vision images in size. Thus, models suffer from heavier memory complexity problems, limiting the scale of the model. Moreover, medical imaging data tend to lack some properties Computer Vision images have (color channels, clear edge features, etc). This can limit the usage of some image augmentation techniques and complicate transfer learning, though these are both efficient tools when working with a limited amount of training data.

Nevertheless, researchers have created different methods to improve DL models and allow them to work with medical images. Especially in segmentation tasks, while each model has its unique characteristics, U-nets[17] and Cascade networks[18] have been the skeleton of most architecture types in the field. U-net uses residual connections across multiple layers of features using different scales to take into account features of various sizes. Cascade networks exclude unnecessary information from the image by first creating a mask using a coarse network, and later using a finer network to get the exact results.

Various brain extraction-specific Deep Learning approaches have been proposed over the years[19–24]. One of the major factors that characterize these models is the dimension the data is treated as. For Deep Learning models in brain extraction tasks, it might be troublesome when trying to train a model that stores all the full 3D features in memory. Most 2D convolutional network methods use 2D slices from all planes to get segmentations from each plane, later merging the results[21,22,25]. Dismantling the 3D image into smaller cubes has also been considered[24]. Other approaches have cropped the image to a smaller size to use it as a form of augmentation as well[19,26]. In our work, we chose to use the 3D image as a whole, decreasing the resolution of the input (EVAC uses a 2mm cube per voxel) to a usable dimension.

Models that work with damaged brains and multiple modalities have also been proposed. BrainMaGe[27] was trained to be more specific to brains with tumors. HD-BET[19] was trained with brains with glioblastoma along with healthy subjects from public datasets to provide a generalized brain extraction method. Both were created to work with any of the 4 structural modalities. Synthstrip[20] created synthesized images by augmenting not just the image itself but on each part of the known whole-head anatomical segmentation. The synthesized images were used to train the model to be generalizable to multiple modalities.

While there are multiple types of architectures proposed in the literature, this paper will focus on one of the most utilized types of models, U-net. The popularity of U-net comes from its clever architecture of forcing the model to learn multi-scale features by using a max pooling layer for every block of layers and using an autoencoder-like structure with skip connections. These skip connections between the encoder and decoder part of the model add an additional pass of information on each scale, making the model more segmentation friendly than classic autoencoders.

Since U-net was introduced, many DL methods that train on medical images now use it as their base architecture[19–22,28]. Though the base model was created for 2D medical image segmentation, there is a wide range of models that modify this architecture to work for 3D[19,29,30]. Other improvements have been introduced as well. Li et al.[31] added cascade networks to U-net to help segmentation on data where the background occupy a major proportion of the image. W-Net[32] used two U-net structures to do unsupervised segmentation of images. HD-BET[19] weighted the complexity of the model to be heavier in the encoder along with loss calculation for each level of the decoder to facilitate training.

## Conditional random fields

CRFs were originally used for segmenting and labeling sequence data[33]. The method is based on assuming the label's probability distribution to be a Markov Random Field (MRF), e.g. affected by only its neighbors. Various approaches have been conducted to use CRFs in image segmentation. The major difference in these methods are (a) the initial segmentation algorithm for the prior distribution (probability of a pixel belonging to a certain label) and (b) how the graph is constructed. Shotton et al.[34] used texture/color/spatial information to create an initial map with four neighboring pixels for the graph structure. Fulkerson et al.[35] created superpixels, a small group of pixels, and calculated the histogram of them to define the initial prior. Enhancing the CRF itself was also suggested using a hierarchical strategy[36,37].

Krähenbühl and Koltun[38] suggested an efficient way of calculating CRFs so that the model could use a fully connected graph instead of just neighboring pairs. Their method uses the position and intensity information of the pixels to create an energy function, which is minimized to reduce the difference between the same labels. The initial segmentation is done by following the feature extraction concept from Shotton et al.[34] We will explain more about the theorem behind their model.

Let us consider $X = X_1, X_2, …, X_N$ to be a vector of label assignments on each pixel 1 to $N$, where the value of each element is within the possibilities of the label. In our case, since brain extraction is a binary segmentation problem, it would be either 0 or 1. Since we are assuming an MRF, it can be approximated as a Gibbs distribution with neighboring interactions. Thus,

given image $I$, we can write a Gibbs distribution $P(X = x|I) = \frac{exp(-E(x|I))}{Z(I)}$. $Z(I)$ is the denominator dependent on only the image itself, so can be ignored. The model aims to find the maximum $x$ that has the highest probability, thus the lowest energy function. Krähenbühl and Koltun[38] defines the Gibbs energy as

$$E(x) = \sum_i \psi_u(x_i) + \sum_{i<j} \psi_p(x_i, x_j) \tag{1}$$

where $\psi_u$ is the unary potential (potential of a single pixel/voxel) and $\psi_p$ is the pairwise potential (potential from its neighboring information). Note that the dependency on the image $I$ was omitted for convenience. In our case, the unary potential will be our base architecture's output. The pairwise potential is defined as

$$\psi_p(x_i, x_j) = \mu(x_i, x_j) \sum_{m=1}^{K} w^{(m)} k^{(m)}(f_i, f_j) \tag{2}$$

where $\mu$ is the compatibility function that serves as a penalty between different labels with similar features and $k^{(m)}$ is the kernel that utilizes feature $f_i, f_j$ information. The paper uses position (smoothness) and intensity/position (appearance) kernels as the feature information. They are defined below.

$$k(f_i, f_j) = w^{(1)} \exp\left(-\frac{|p_i - p_j|^2}{2\theta_\alpha^2} - \frac{|I_i - I_j|^2}{2\theta_\beta^2}\right) + w^{(2)} \exp\left(-\frac{|p_i - p_j|^2}{2\theta_\gamma^2}\right) \tag{3}$$

$p_i$ and $I_i$ are each the spatial and intensity information of pixel i. $\theta_\alpha, \theta_\beta, \theta_\gamma$ controls the degree of each feature.

Krähenbühl and Koltun[38] also provide a way of efficiently approximating the posterior distribution (essentially the probability map) using mean field approximation. A quick summary of the method is described here. It assumes an independent relationship between the latent variables, in our case the true labels of each pixel/voxel. This creates an optimizable approximation of the true posterior. Our goal is to find the closest approximation of the posterior, the true label distribution. Thus, we aim to minimize the KL-Divergence, a measure of distance between two distributions, between the posterior and its approximation. Through Bayesian inference, one can calculate that the prior is a sum of the KL-Divergence and negative variational free energy. As mentioned in the previous paragraph, Krähenbühl and Koltun[38] showed that this can be written in the form of the unary potential and the pairwise potential. The assumption on the independence of the latent variables lets each variable be updated separately treating the other variables and the prior as constant. Thus, each iteration of the latent variable update that increases the negative variational free energy is guaranteed to decrease the KL-Divergence. Optimally, the converged output would be the true segmentation probability map of the image. More details on the method can be found in Krähenbühl and Koltun[38].

The Deep Learning model's image segmentation output is mainly a softmax result of the final layer. Since this gives a probability distribution of a pixel belonging to a certain label, using CRFs after a Deep Learning model would be a good way of improving the final result beyond the base model. CRFasRNN[39] integrated the iteration process from Krähenbühl and Koltun[38] as a Recurrent Layer. Unlike the original, fully connected CRF model including the CRF process in the model architecture itself increases optimization efficiency and reduces the number of hyperparameters. Since then, there have been suggestions to improve the model[40], but not for volumetric medical images. While both the manual and Recurrent Layer approaches have been utilized in many medical imaging domains[41–46], it has been suggested that using naive CRFasRNN layer within the Deep Learning architecture does not help much with the volumetric segmentation[47].

The reasoning behind this lack of performance could be several reasons, but in this work, we focus on the single-channel problem. When CRFs are used in Computer Vision, the feature space is constructed using color channels. However, many medical images do not have multiple channels. Thus, the CRF algorithm will not have enough information to refine the initial segmentation output.

## Summary of our work

This paper aims to improve V-net[30], a variant of U-net[17] for 3D medical image segmentation. V-net has several advantages compared to U-net, which include residual layers, replacing pooling layers with convolutional layers and utilizing Dice Loss instead of Binary Crossentropy. While most recent U-nets do share the characteristics that V-net proposed, V-net is specifically calibrated towards volumetric medical image segmentation. Using V-net as base architecture, we focus on two things which, if solved, can greatly improve its performance in brain extraction. First, the lower scale layer's information is highly dependent on features from a higher scale. Thus, at the beginning of a training phase, lower layers will not have adequate information to process. This inefficiency of training is especially problematic in brain extraction tasks since training speed is slow due to the limited mini-batch size. Second, imperfect training data labels can introduce unnecessary biases or errors for the model to learn, limiting the model's generality. Our work suggests three ways to resolve these problems: (1) multi-scale inputs, (2) a unique utilization of the Conditional Random Fields as a Recurrent Layer (CRFasRNN)[39] and (3) a loss function to further remove potential errors introduced by non-DL methods that were used in creating training labels.

Results indicate our model's high performance in multiple test datasets, both quantitatively and qualitatively. The performance is best highlighted in how it provides the most accurate boundaries along the brain mask. We additionally include prediction masks of unseen non-adult or patient T1 images to emphasize the generalizability of our model. Validation loss plot during training indicates that our proposed approach also increases training efficiency.

## Methods

Due to the size and complexity of brain MRI and the lack of accurate public data, it is incredibly important to have a robust and efficient model that works with limited resources. This paper will propose three ways to enhance Deep Learning architectures for brain extraction: (1) multi-scale inputs for the efficiency of the model, (2) a distinctive usage of the Conditional Random Field Recurrent Layer and (3) a matching loss function to get a finer segmentation result. The overall architecture is shown in Fig. 1. We would like to emphasize that all of these approaches have minimal to no effect (ranging up to 0.3% change) in the number of parameters. To be more specific, the base V-net model has 27,008,316 parameters, the multi-scale input adds 88,032 parameters and the CRF layer adds 32,052.

### Multi-scale

We first propose feeding raw multi-scale inputs to the model, as previously suggested by other segmentation models[48,49]. We add the lower-scale inputs by concatenating them to the output of each downsampling convolutional layer as shown in Fig. 1. We revisit the reasoning behind this as it has a synergistic effect with our other improvements.

We can write a general equation for the output of a layer in the model as

$$y = \sigma(z) \tag{4}$$

$$z = wx + b \tag{5}$$

where $w$ is the weight, $x$ is the input, $b$ is the bias and $\sigma$ is any non-linear function. The gradient of $w$ is calculated through backpropagation to update the weight (kernel in convolutional layers). By the chain rule, we know that the gradient value is a multiplication of the gradients of the deeper layers, the non-linear function and the

input to the current layer. We can write this as

$$\frac{\delta L}{\delta w} = G \frac{\delta x'}{\delta z} \frac{\delta z}{\delta w} \qquad (6)$$

where $L$ is the loss function, $G$ is the previous gradients in the chain rule, $x'$ is the input to the next layer. There are two major changes one can make in the layer to modify the target gradient. One is altering the non-linear function, hence changing $\frac{\delta x'}{\delta z}$. We aim to change $\frac{\delta z}{\delta w}$ which is the input to the layer. In other words, we are defining $x$ to be

$$x = x_{prev} + x_{raw} \qquad (7)$$

$x_{prev}$ stands for the input from the previous layer that reduces the feature size. This is present in any U-net-type architecture. $x_{raw}$ is the raw input resized to match the layer's feature shape. It forces the weight to update using meaningful information even if the $x_{prev}$ provides a non-important feature, especially when training from a randomly initialized model.

The lower scale inputs will ultimately reinforce the multi-scale scheme of V-net so that each end of the encoder structure will contain more accurate corresponding scale features. We utilize this in our next proposed change.

## Enhanced CRFasRNN

While the multi-scale inputs provide good results alone, they can still contain some bias from the non-human-made prior labels. We need our output to follow the complex, intricate and detailed structure of the brain, which is often not captured in the ground truth. Many features are considered when calculating a brain mask, but one cannot deny the fact that continuity in intensity and structure is an important factor. As CRFs are known to minimize the discrepancies in labels between neighboring pixels with similar intensity and spatial information, they can be used to create a fine-grained segmentation result.

Previous work have shown the effectiveness of the CRF method as a post-processing step after the deep-learning step[41]. However, the number of hyperparameters of the method can limit the flexibility of the overall model. Hence, as implemented in Monteiro et al.[47], we add a Recurrent Layer that

essentially does the matrix factorization optimization iteration step of the CRF. The layer is added right after the output of the multi-scale model part of the network.

Two ways of training were tested. (1) Training the whole model from scratch. (2) Pre-training and fixing the weights of the base architecture, followed by training the CRF layer. Though fixing the weights can prevent any unnecessary change in the optimized weights of the base model, training all the weights from scratch provided better results. This is possibly due to the fact that the previous model already learned a stable local minimum. This could have limited the CRF layer's ability to correct the output. We used CRFasRNN Layer as provided in Monteiro et al.[47] and Adams et al.[50], which provides an efficient way of calculating the kernel features. Instead of 10 iterations recommended in the previous studies, we found 20 iterations to give much better results, which could be decreased back to fewer iterations during prediction as suggested in Zheng et al.[39].

However, this does not solve the single-channel problem of gray-scale volumetric medical images. Thus, one of our major contributions is adding learned features within the network along with the image intensity to improve the CRF layer. Specifically, we used the last layer of the highest scale output of the encoder in the model. While our ablation study shows that it alone has a beneficial effect on the model's performance, the more accurate features created from the multi-scale inputs have a synergistic effect with our modified CRFasRNN layer.

## Loss for CRFasRNN

While in practice, we also encountered problems where sometimes the CRFasRNN layers fail to create a meaningful change in the model output, even when trained together from scratch, on certain areas more stable intensity-wise than the others. Hence, we propose adding a second loss function, negative Dice Loss between the original model's output and the CRF layer's output. Given a good loss weight, this can enforce the base model to give a more stable crude segmentation while the CRFasRNNlayer improves the segmentation result by a reasonable amount. We can write the proposed loss function as this

$$D(y, y') - \lambda D(z, y) \qquad (8)$$

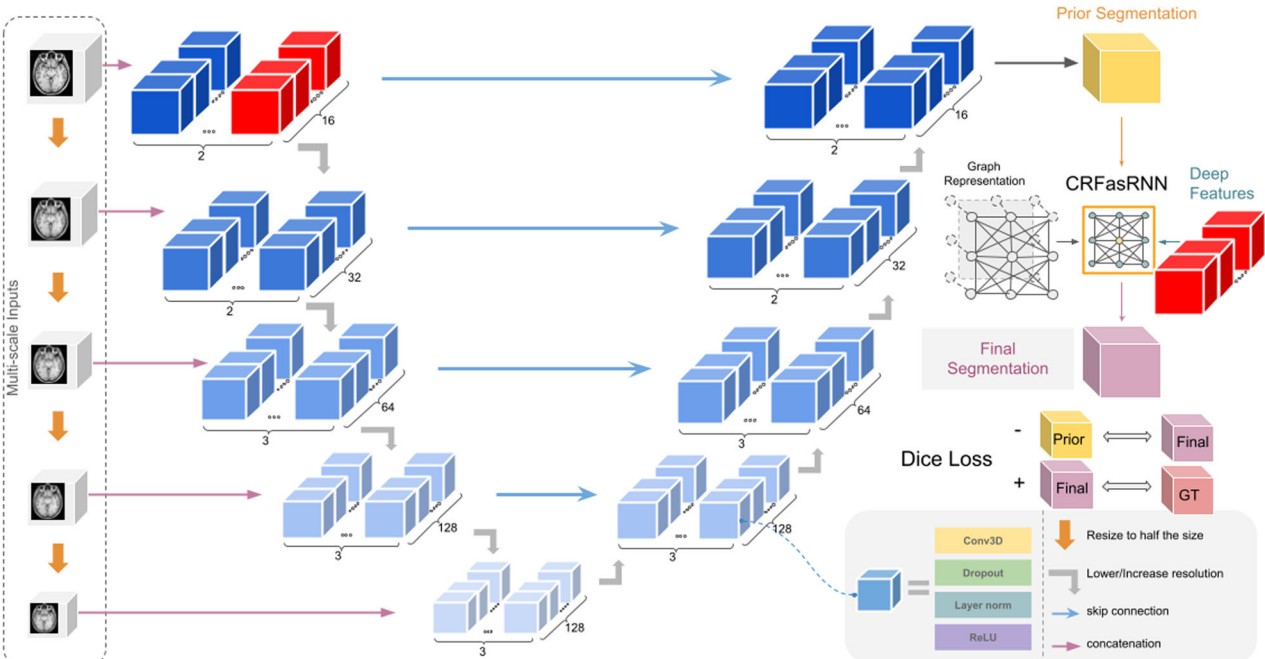

**Fig. 1 | Summary of the model architecture.** The architecture uses V-net as its base model with the following important changes: multi-resolutional raw inputs, modified CRFasRNN and additional Dice Loss. The CRF layer uses the rear layer of the first level of the encoder. A negative Dice Loss is calculated between the base model's output and CRF layer's output.

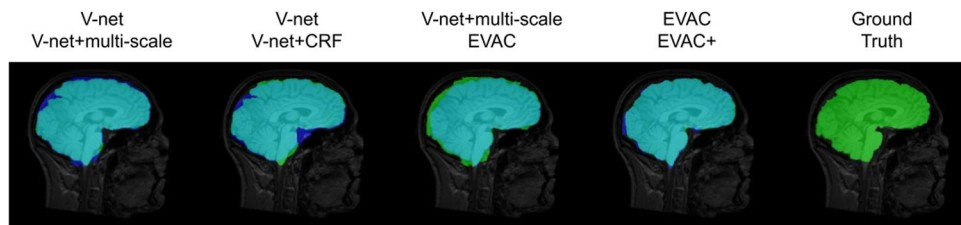

**Fig. 2 | Qualitative ablation study.** Blue represents a model without a certain change, green the model with one of the proposed changes and teal the overlapping regions. Ground truth is added for reference. We can see that while both multi-scale inputs and conditional random fields do a great job of recovering False Negatives and removing False Positives, a combination of them gives a better segmentation. The additional loss function also fine-tunes the output closer to the ground truth.

Where $D$ is Dice Loss $y$ the final prediction, $y'$ the provided ground truth, $\lambda$ the additional loss weight and $z$ the prediction before the Recurrent layer.

**Additional processing.** For all the methods tested, including our model, we have removed small segmentation errors that might be present in the result. Using prior information about the brain structure, we have limited the segmentation result to the largest connected component, while filling the black spaces within the segmentation.

### Training and testing details
Our models were trained using two Tesla V100s. All the models compared in the ablation study were trained until convergence. More specifically, EVAC, EVAC+, V-net with CRF layer, V-net with multi-scale inputs and base V-net were respectively trained for 18, 18, 18, 32, 40 epochs. A learning rate of 0.01 was used. Dropout rate of 0.2 was used for the initial layer and 0.5 was used for the rest.

1438 T1 MR images were used for training from the following public datasets: the Human Connectome Project(HCP)[16], CC359[15] and NFBS[14]. The provided labels of the datasets were used for training. All training data used were healthy adults, without any known abnormalities. CC359 and NFBS had a resolution of 1 mm³ while data from HCP had 0.7 mm³. 10% of the training data was used for validating the model's performance. For testing purposes, 39 images from the LPBA40 dataset[51] and 30 images from the Hammers dataset[52,53] were used. Both datasets contain healthy adult brains that were not seen by the model during training. To show the performance as images, IXI dataset[54] was used as healthy adult qualitative analysis data. Clinical (Alzheimer and Parkinson) and pediatric data from OASIS3[13], FCP-INDI[55] and HBN[56] were used to test the robustness. Note that unlike many state-of-the-art methods, it was trained only on public datasets of healthy adults with minimal augmentation sampling, described below. More details on the datasets are available from each corresponding paper. DIPY Horizon[57] was used for visualization.

Each image was loaded to a common space of 1mm cube per voxel using the affine matrix accompanied by the file. The image was translated so that the center of the image would be the voxel coordinate of (128, 128, 128). The image was padded or cropped to have a size of (256, 256, 256). No additional registration to the template process was required.

Augmentation has been known to force the model to learn structural components. It is particularly important for medical images due to the lack of data. The image was first normalized to a range of 0 to 1. For intensity augmentation, we have used scaling (range of 0.9 to 1.1) and shifting (range of -0.1 to 0.1) of values to add/multiply random noise. For transformation augmentation, random rotation (maximum 15 degrees) and translation (maximum range of 10mm) were used. This was to compensate for most of the minor differences T1 weighted images might have when in the same coordinate space. On every epoch, a single random augmentation method was chosen for each image. We only trained on the augmented images, i.e. did not increase the size of the training dataset per epoch. This was done to improve generality without increasing the time complexity of training. We would like to emphasize EVAC+ was able to work with the more extreme range of augmentations while models without the modified CRFasRNN layer could not.

More details on the training environment and the exact parameters for the model architecture can be found in our code https://github.com/pjsjongsung/EVAC.

### Statistics and reproducibility
The Dice Coefficient and Jaccard Index were calculated by replicating the equations using the Python package Numpy[58]. Scikit-image[59] python package was used for calculating Hausdorff Distance. All models were run through the default parameters given in their corresponding software or repository to collect their predicted masks. P values were calculated using Wilconxon's method implemented in the Python package Scipy[60].

### Ethics
Institutional Review Board approval was waived for our study because we used publicly available datasets.

### Reporting summary
Further information on research design is available in the Nature Portfolio Reporting Summary linked to this article.

## Results
Evaluation between models was done with three metrics. The Dice Coefficient and Jaccard Index were selected to measure the similarity of the prediction to the ground truth. While both are used for the evaluation of image segmentation, the Dice Coefficient adds more weight to the True Positives than the Jaccard Index. Hausdorff Distance was used to measure the surface distance error. Higher scores are better for the Dice Coefficient and Jaccard Index. For Hausdorff Distance, lesser scores are better.

In the figure descriptions, EVAC and EVAC+ refers to our model without and with the additional loss. V-net+multi refers to a V-net model with the multi-resolution scheme and V-net+CRF refers to V-net with the CRFasRNNLayer but without the multi-scale inputs.

### Ablation study
We first compare results from V-net[30], V-net with multi-scale inputs or CRFasRNN layer, EVAC, and EVAC+ with the proposed loss function to show the development of results per each change in the model. Figure 2 provides a summary of what each enhancement to the model does in terms of output. Ground truth image is also provided in the figure for overall performance evaluation.

Each improvement (multi-scale input, the proposed CRF layer and loss function) of the architecture corresponds to an improvement of accuracy. The multi-scale input enforces the model to use large-scale features, which is missed in the V-net output. However, this also introduces non-brain regions to the segmentation result. The proposed CRF layer corrects most of the false positives. However, it is not completely free from the base model having major influence over the CRF layer. We can see that the proposed change in loss function projects additional importance to the CRF layer, inducing greater correction in the segmentation output. To show that the proposed CRF layer is beneficial even without the multi-scale scheme, we compare V-net like model with the same model plus the CRF layer. You can see that it

oversegments some regions, but recovers important brain regions as expected.

The improvement is not only in the accuracy but the efficiency of training. Figure 3 shows how the loss changes with the number of epochs. Note that the training environment is identical between models except the proposed changes (e.g. CRF layers, additional loss). The plot clearly shows the learning efficiency increase per enhancement of the model. Another important phenomenon is that even though the regularizing Dice Loss is applied at the model level, it still leads to faster convergence of the original loss.

We further show integrated gradients[61] to provide an explanation of how our changes are improving the model. Figure 4 shows that our changes lead the model to focus less on unimportant regions within the image. Comparison between EVAC and V-net+multi-scale suggests that our proposed CRF layer pushes the model toward better usage of multi-scale inputs. The difference in feature importance within the brain between EVAC and EVAC+ is possibly due to the CRF layer's refining step playing a major role, thus needing less focus on low-level features. This can be beneficial when dealing with abnormal or noisy images.

Quantitative results on the test dataset is shown in Table 1 to emphasize the synergistic effect. While the proposed changes individually do not help the model's accuracy, EVAC and EVAC+ in the end manage to get lower Hausdorff Distance, indicating that the proposed changes have a positive effect in refining the cortical surface regions.

We would like to emphasize that besides some minor details (e.g. number of epochs until convergence) the models were unchanged except for the major proposed changes. Thus, the increase in accuracy would have been purely from the improvement of the model architecture.

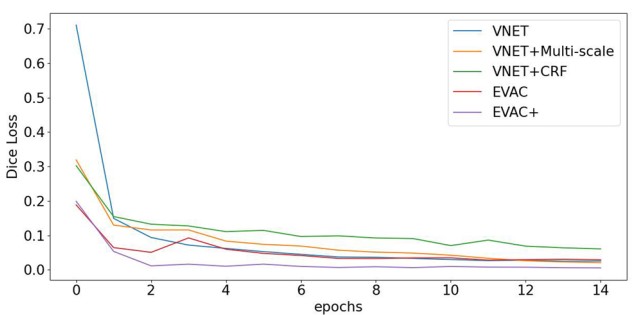

**Fig. 3 | Dice Loss comparison for ablation study.** The original Dice Loss plot. The values in the plot do not include the proposed regularizing negative Dice Loss of our model. It clearly shows an efficiency increase in training for each improvement. Note that our model (EVAC+) with the additional Dice Loss trains better even in terms of the original Dice Loss.

We also show Fig. 5 to emphasize the robustness of our methods. The images are from the OASIS3 dataset[13], Parkinson's from FCP-INDI[55] and pediatric brains from the Healthy Brain Network dataset[56]. It is evident that our method is robust in a wide area of clinical/pediatric data, even though such data were never introduced during training.

## Comparison

We compare our EVAC model to other publicly available state-of-the-art methods. The models were chosen based on the accessibility. For Deep Learning models, only pre-trained models for brain extraction were chosen for two reasons. First, the training dataset is a critical part of DL methods in brain extraction. Thus, it is not reasonable to train the models again with a fixed dataset. Also, since we are planning to release the pre-trained model for actual use, it is more reasonable to compare to the models that are provided for the same purpose.

Figure 6 shows the results of each model on an image from the IXI dataset. Both the EVAC model with and without the proposed loss function were included to emphasize its effects. While most methods fail to remove the dura mater on the surfaces of the brain or over-segment, both of our models are mostly free from that problem. BEaST is another method that is free from this issue but is known to have problems around the cerebellum, which our models do not. Additionally, by using the additional loss function, our model returns segmentation outputs with higher detail.

Figure 7 shows the average of each metric on the test datasets as box plots. Our method has a stable accuracy in both datasets. Table 2 shows the significance of the difference between the averages through $p$ values. The values were calculated through Wilcoxon's signed rank test method. Red shows cases where our method has a higher average score than the compared method. Gray indicates insignificant p values.

## Discussion

Our model provides a robust and efficient DL model for brain extraction with limited data and augmentations. Results clearly show the stable accuracy compared to other methods, especially in the cortex/dura mater interface. The ablation study performed suggests that Deep Feature CRFasRNN layer with multi-scale inputs and negative Dice Loss have synergistic effects. The improvement can be seen not only in accuracy but also in efficiency during training.

Despite our model's performance in the evaluation dataset, it is limited to T1 weighted MRI data unlike other proposed methods[19,20,27]. However, we believe when provided with enough variety in images, the model would be able to train up to a state where it can provide output for abnormal brains, as the model utilizes features from the image, which is more robust to the type of image. The additional loss would also help correct local issues with the segmentation. This is evident in the robustness analysis results, where our model succeeds in brain extraction of clinical or pediatric brains. The results

**Fig. 4 | Important image features highlighted through integrated gradients.** Larger value indicates a larger gradient within that region, suggesting higher importance in prediction. Note that our model is not affected by most non-brain regions such as dura mater and neck regions. The figure also shows that EVAC+ has less important features within the brain, suggesting that the proposed CRF layer is contributing more to the model.

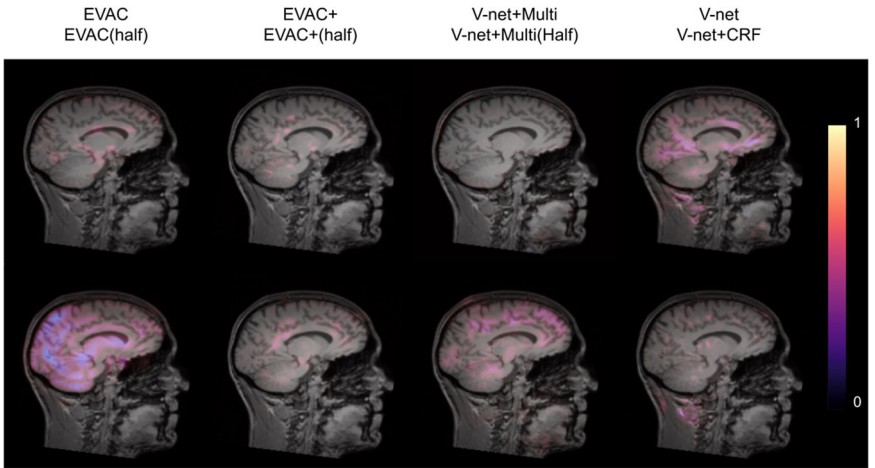

**Table 1 | Quantitative ablation study comparing metrics (mean ± standard deviation) between models with different levels of change in the architecture[a]**

| Model type | LPBA40 ($n$ = 39) | | | Hammers ($n$ = 30) | | |
|---|---|---|---|---|---|---|
| | Dice Score | Jaccard Index | Hausdorff distance | Dice Score | Jaccard Index | Hausdorff distance |
| V-net | 0.964 ± 0.007 | 0.931 ± 0.013 | 12.5 ± 5.15 | 0.928 ± 0.011 | 0.866 ± 0.019 | 14.3 ± 3.20 |
| V-net + MS[b] | 0.948 ± 0.010 | 0.901 ± 0.017 | 54.0 ± 14.5 | 0.927 ± 0.014 | 0.865 ± 0.024 | 13.3 ± 2.50 |
| V-net + CRF | 0.913 ± 0.020 | 0.840 ± 0.033 | 51.4 ± 10.3 | 0.920 ± 0.014 | 0.865 ± 0.024 | 13.3 ± 2.50 |
| EVAC | 0.958 ± 0.007 | 0.920 ± 0.012 | 11.4 ± 4.61 | 0.940 ± 0.005 | 0.887 ± 0.009 | 9.78 ± 2.43 |
| EVAC+ | 0.958 ± 0.006 | 0.919 ± 0.011 | 9.54 ± 4.59 | 0.947 ± 0.006 | 0.900 ± 0.011 | 8.75 ± 2.16 |

[a]LPBA40 and Hammers used as test datasets.
[b]Multiscale.

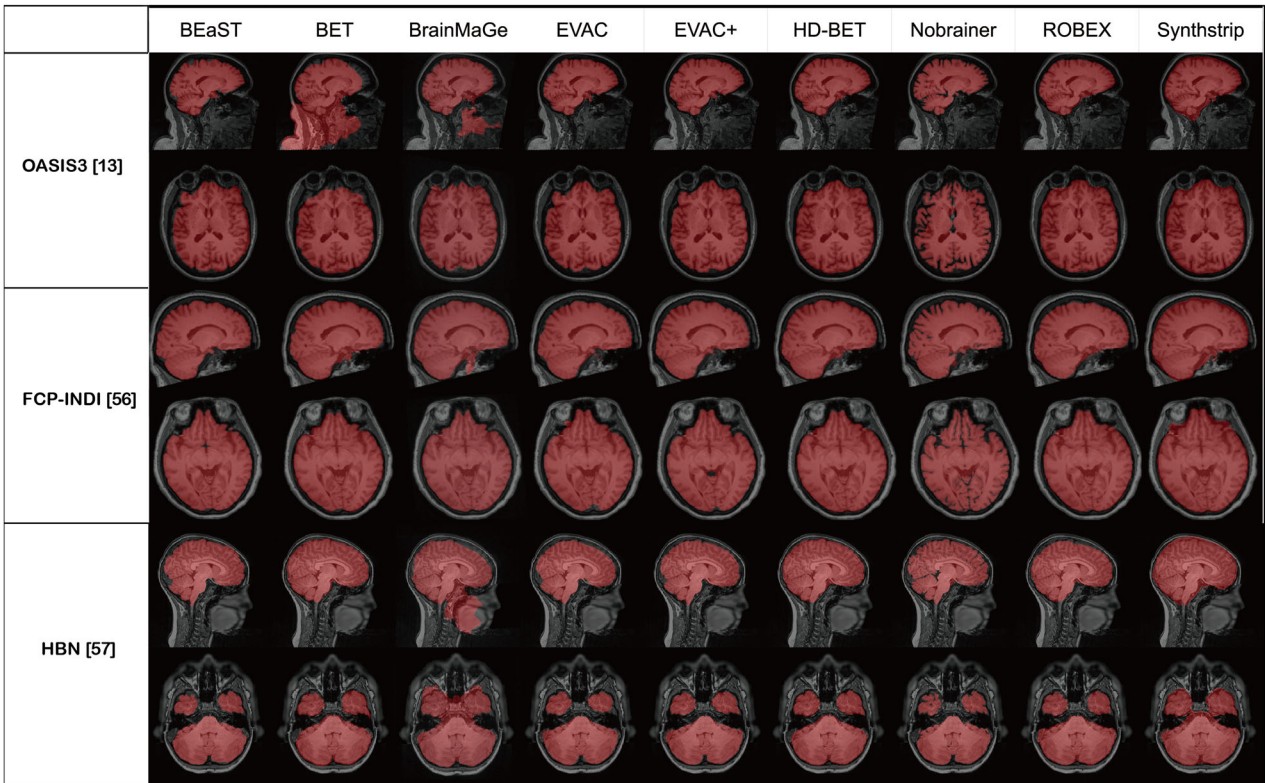

**Fig. 5 | Qualitative robustness analysis of the compared methods.** The images are from OASIS, Parkinson's dataset from the FCP-INDI project and pediatric brains from the Healthy Brain Network dataset. Red marks the segmentation results of each method.

are important because these populations were not included in the training dataset. This suggests the model's capability to train and perform on limited and biased data, which is one of the major issues in the medical imaging field. The model can be, however, negatively affected by artifacts that are not evident by large intensity differences (e.g. artifacts in the skull connected to the brain structure with similar intensity). This can only be avoided by anatomical priors (e.g. overall atlas shape of the brain). Further work would be extending the dataset for general use by adding more variety in modalities or clinical cases of the training data and balancing between feature-based segmentation refinement and anatomical priors.

Our model provides a precise semantic segmentation output near the outer surface of the brain. This can be beneficial in any further analysis or clinical steps that are highly dependent on the boundaries of the brain (e.g. cortical thickness). For example, due to the difference in the portion of gray matter and white matter, methods trained on adult brains can easily over or under-segment pediatric brain images, leading to errors in tractography or surgical procedures.

This work limits the model's goal to T1 brain extraction task. However, in theory, the same model can be trained with other modalities. In addition, a wide range of medical images share the characteristic that neighboring regions have fairly consistent features. Therefore, while not currently tested, the proposed changes might be beneficial in a variety of medical image tasks. Another direction the model can take is moving from brain extraction to general-purpose medical image segmentation (e.g. multi-label). In principle, this version should be compared with methods such as nn-Unet[28] and Cotr[62].

For the past few years, BEaST had the top accuracy among both traditional and Deep Learning methods. However, the time complexity of the method has prevented it from being widely used in the field. Thus, even though it is often problematic, BET has been the top-used Skullstripping method. Our method however clearly shows comparable or higher accuracy and does not suffer from such problems as it is a Deep Learning model. Therefore, we believe our method could be not just for the medical image segmentation community but all fields that could benefit from an accurate Brain Extraction.

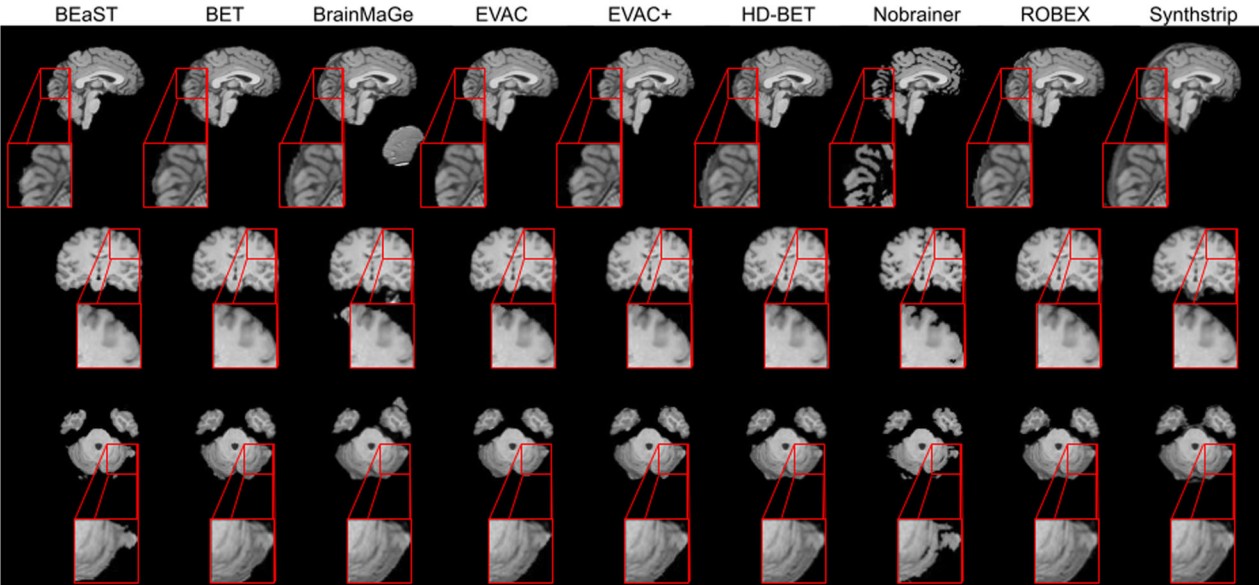

**Fig. 6 | Qualitative comparison between models.** The image shows comparison between our models with (EVAC+) and without the additional Dice Loss (EVAC) and other established state-of-the-art models. Specific regions were zoomed in to emphasize the improvements our model is achieving. Results show that our models get an accurate local segmentation on the surface, whereas most of the other methods either under-segment or include the dura mater. Similar accuracy improvements are also visible near the central sulcus and the cerebellum. T1 images from the IXI dataset.

**Fig. 7 | Quantitative analysis between models.** The plots indicate (**a**) Dice Coefficient, (**c**) Jaccard Index, and (**e**) Hausdorff Distance for the LPBA40 dataset ($n = 39$) and (**b**) Dice Coefficient, (**d**) Jaccard Index and (**f**) Hausdorff Distance for the Hammers Atlas dataset ($n = 30$). Our models with (EVAC+) and without the additional Dice Loss (EVAC) have a stable near-top accuracy in both datasets and metrics, while others have either lower scores in a dataset or unstable results.

**Table 2 | P-values from the quantitative comparison between models calculated using Wilcoxon's signed rank test method[a,b]**

| Model compared against EVAC+ | LPBA40 (n = 39) | | | Hammers (n = 30 | | |
|---|---|---|---|---|---|---|
| | Dice Score | Jaccard Index | Hausdorff distance | Dice Score | Jaccard Index | Hausdorff distance |
| BEaST | **5.26e−08** | **5.26e−08** | **1.80e−07** | 9.27e−03 | 9.27e−03 | 1.98e−01[*] |
| BET | 0.01e−02 | 9.88e−05 | **1.11e−06** | **1.73e−06** | **1.73e−06** | **8.94e−04** |
| BrainMaGe | **5.26e−08** | **5.26e−08** | **5.26e−08** | **1.73e−06** | **1.73e−06** | **1.73e−06** |
| EVAC | 1.20e−02 | 1.15e−02 | **9.29e−05** | **1.73e−06** | **1.73e−06** | **1.57e−02** |
| HD-BET | 5.68e−08 | 5.68e−08 | **4.08e−05** | **6.98e−06** | **6.98e−06** | 1.44e−01[*] |
| Nobrainer | **5.26e−08** | **5.26e−08** | **1.78e−04** | **1.73e−6** | **1.73e−06** | **2.35e−6** |
| ROBEX | **5.26e−08** | **5.26e−08** | **5.26e−08** | **1.73e−06** | **1.73e−06** | **2.84e−05** |
| Synthstrip | **9.83e−03** | **1.07e−2** | **1.25e−03** | **1.73e−06** | **1.73e−06** | **1.73e−06** |

[a]LPBA40 and Hammers used as test datasets.
[b]Bold values indicate comparisons where EVAC+ had a superior average value
[*]Insignificant p values.

Medical image processing has developed rapidly since DL was first introduced in the field. Despite the common elements between using DL for computer vision images and medical images, crucial factors such as the quality/quantity of training data, different weights on false positives/negatives and bias in labels have pushed the field to have more carefully and ethically designed methods. For instance, techniques such as One or Few shot learning[63,64] fine-tune a pre-trained model with a limited number of data. Federated learning[65] proposes indirectly using non-public datasets to improve the generalizability of the method while not raising privacy concerns. Tversky Loss[66] has been helpful in solving some of these problems by weighting Dice Loss based on the portion of foreground and background segmentation. We aim to contribute to this field by introducing the robustness of our model to bias and unseen data. Brain extraction is a mandatory pre-processing step for a majority of methods. Over or under segmentation of the brain can be one of the leading factors towards an inaccurate analysis. Thus, our model's robustness is critical in numerous clinical and research tasks.

The paper emphasizes the enhancement of the overall architecture, but keeps the changes in the base architecture minimal. However, there are various changes we can make to the base V-net architecture. Instance Normalization was proposed in StyleGAN[67] and had cases where it performed reasonably well in medical imaging tasks[68]. Many losses have been proposed as alternatives to Dice Loss[20,66]. Even though we are not using the whole Vision Transformer[11] architecture, it has been reported that just applying self-attention to the Convolutional Neural Networks could be beneficial to its robustness[69]. Future work would be refining our model with these techniques for better quality outputs.

It is still controversial on whether a segmentation model should be based on Vision Transformers or Convolutional Networks. Despite the recent advancements in Vision Transformers, we did not focus on or use the method in this paper as they tend to be trained on more training data and training epochs[70], which is often not suitable for large volumetric medical images. Also, despite the efforts to combine U-net and Transformers to utilize more global contextual info[71], our model focuses more on correcting local errors, which is where the errors from methods are in Brain extraction tasks. Thus, it relies on local contextual info. Using CRF layers on global info with Transformers needs more investigation.

While CRFasRNN has lower hyperparameters compared to the original CRF model, it can still increase the complexity of choosing an ideal model for a specific task. In our work, we pre-process the images so that the voxel resolutions of the training and test dataset are equal or resized to be equal. Though this does not completely remove the need to tune the hyperparameters, it should minimize the parameter space. More investigation should be done on different segmentation tasks.

Despite the fast speed of the Deep Learning prediction using GPUs, CRFasRNNLayer with more features can still be relatively time consuming (About a 2.5-fold increase in GPU calculation time). One possible workaround can be reducing the time complexity by using fewer iterations within the CRF layer as the original paper suggests. However, training simpler models with the highly accurate predictions from the original model can completely remove the time-consuming step during inference. This can also solve some issues with implementing the model since the original code has a sub-optimal CPU implementation and recent package support due to compiler changes. Future work would be retraining the outputs of the model with a simpler model to keep the benefits of the CRF layer's output refinement while reducing time complexity.

The Deep Learning in medical imaging community has developed various ways to solve problems such as bias and transparency of the training data. Public datasets for several different tasks are being released[72,73], open-source packages such as MONAI[74] are pushing federated learning (training across sites without directly accessing private patient data) and transparent models. We envision our addition to the field by introducing a flexible model that can overcome bias in training data and perform on unseen data as well could further improve the field, reducing dangers of wrong and biased predictions and privacy concerns from requiring numerous data.

Removing non-brain tissues from MRI data is a necessary step for nearly all forthcoming analyses of these images. Various Deep-learning approaches have been considered in the literature with limited outcomes. To resolve this issue, we propose EVAC+, a DL architecture that combines a V-net architecture with CRF layers and multi-scale inputs. Results show that the proposed CRF layer with a negative Dice Loss for refining the segmentation can significantly improve results. The improvements are especially profound in cortical surfaces, which most methods fail to achieve a precise segmentation. The accuracy is also profound in clinical data and pediatric brains, even though those were not included in the training data. We additionally show that this ready-to-go approach can efficiently reach higher accuracy with fewer epochs. We believe it is exciting and refreshing that a revisit to CRFs in medical imaging can be used in tandem, creating a compound effect with Neural Networks, and further improving crucial medical imaging problems. A version of our model is currently open-sourced via DIPY[57] to disseminate its use to the wider medical imaging community via a stable platform.

## Data availability

We used public datasets available online. The following datasets could be accessed at the stated url: **NFBS** http://preprocessed-connectomes-project.org/NFB_skullstripped/[14], **CC359** https://sites.google.com/view/calgary-campinas-dataset/home[15], **FCP-INDI** https://fcon_1000.projects.nitrc.org/[55], **HBN** https://github.com/richford/hbn-pod2-qc[56], **IXI** https://brain-development.org/ixi-dataset/[54], **Hammers** https://brain-development.org/brain-atlases/[52,53], **LPBA40** https://www.loni.usc.edu/research/atlas_downloads[51] The remaining datasets were accessed by registering an account and/or agreeing and following the stipulated requirements to use the requested dataset: **HCP** https://www.humanconnectome.org/study/hcp-

young-adult/document/1200-subjects-data-release[16], **OASIS3** https://www.oasis-brains.org/[13] Comparison data used in creating the plots are provided in this link: https://figshare.com/articles/dataset/Metric_scores_used_in_b_Multi-scale_V-net_architecture_with_deep_feature_CRF_layers_for_brain_extraction_b_/24463627[75]. Any image data can be recreated by gathering predicted masks from the available code or the corresponding model's software/repository.

## Code availability

Code and tutorials for EVAC+ are publicly available in GitHub and can be found here https://github.com/pjsjongsung/EVAC[76]. Code is written using Python language with dependencies in DIPY[57] and Tensorflow[77]. The CRFasRNN layer skeleton was adapted from https://github.com/MiguelMonteiro/CRFasRNNLayer. A version of EVAC+ is publicly available through DIPY[57] at https://github.com/dipy/dipy.

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

## Acknowledgements

The authors would like to acknowledge that the research reported in this publication was supported by the National Institute Of Biomedical Imaging And Bioengineering of the National Institutes of Health under Award Numbers R01EB027585 and 2R01EB017230-05A1. We are also grateful to Indiana University for providing access to computing clusters Bigred200 and Carbonate systems with NVIDIA/TESLA GPUs for training and testing our method. Finally, we would also like to thank the Graduate Program in Intelligent Systems Engineering and Program in Neuroscience of Indiana University, Bloomington for sponsoring Jong Sung Park.

## Author contributions

J.S.P.: Conceptualization, Collecting public data, Planning model architecture, Model experimentation, Statistical analysis, Writing the first draft, Editing and Review S.F.: Conceptualization, Editing and review E.G.: Conceptualization, Planning model architecture, Editing and review.

## Competing interests

The authors declare no competing interests.
