## [Peer Review File · Communications Medicine]

Reviewers' comments:

Reviewer #1 (Remarks to the Author):

The paper discusses the importance of brain extraction in brain imaging research and how the complex structure of the interfaces between the brain, meninges, and human skull makes it challenging to find a highly robust solution. With the advent of deep learning, the authors propose a novel architecture called EVAC+ that improves training efficiency, accuracy, and segmentation output. They compare their model to state-of-the-art non-DL and DL methods and show that EVAC+ outperforms in most cases, achieving a high and stable Dice Coefficient and Jaccard Index with a desirable lower Surface (Hausdorff) Distance. The authors also show that their approach accurately segmented clinical and pediatric data, despite the training dataset only containing healthy adults. The model is publicly available and open-source, which can benefit a wide range of researchers.

1. Expand the model name in the first time usage for EVAC and EVAC+
2. Authors can explain why they choose V-Net as their base architecture. What are its pros and cons compared to UNET
3. In line 175, author discussing that there are only few methods that use 3D UNET for medical image segmentation (analysis), which is not true. There are plethora of 3D UNETS available for medical image segmentation, classification and synthesis.
4. Adding multi scale inputs should have increased memory load on the model. Authors can discuss the computational efficiency of the models compared too.

Reviewer #2 (Remarks to the Author):

This paper presents a EVAC+ network for brain extraction. EVAC+ has three innovations: a smart augmentation method to improve training efficiency, a CRF recurrent layer to improve accuracy and a loss function to fine tune the segmentation output. The authors demonstrated the effectiveness of the proposed approach. However, the techniques of this paper is very limited. My detailed comments are below:

1. This paper aims to enhance V-net performance by incorporating multiple scale inputs, CRFsRNN, and a loss function to mitigate potential errors introduced by deep learning methods. While prior research such as Cotr[1] and nnunet [2] have shown superior performance for 3D medical image segmentation, the authors have not mentioned or compared their approach with these methods.
[1] Xie, Y., Zhang, J., Shen, C., & Xia, Y. (2021). Cotr: Efficiently bridging cnn and transformer for 3d medical image segmentation. In Medical Image Computing and Computer Assisted Intervention–MICCAI 2021: 24th International Conference, Strasbourg, France, September 27–October 1, 2021, Proceedings, Part III 24 (pp. 171-180). Springer International Publishing.
[2] Isensee, F., Jaeger, P. F., Kohl, S. A., Petersen, J., & Maier-Hein, K. H. (2021). nnU-Net: a self-configuring method for deep learning-based biomedical image segmentation. *Nature methods*, 18(2), 203-211.
2. In terms of novelties such as multi-scale inputs, CRF and a loss function to fine tune segmentation output, these ideas are very common in this field.
[3] Kamnitsas, Konstantinos, Christian Ledig, Virginia FJ Newcombe, Joanna P. Simpson, Andrew D. Kane, David K. Menon, Daniel Rueckert, and Ben Glocker. "Efficient multi-scale 3D CNN with fully connected CRF for accurate brain lesion segmentation." *Medical image analysis* 36 (2017): 61-78.

Reviewer #3 (Remarks to the Author):

Key Results: The study provides a detailed evaluation of a novel method for brain extraction in medical imaging using deep learning. The proposed method, EVAC+, demonstrates improved performance over existing state-of-the-art methods, as evidenced by high Dice Coefficient and Jaccard Index scores, as well as lower Hausdorff Distance. The study utilizes CRF as RNN to the network to achieve the results.

Validity: The authors propose their novel method and apply it to the task of brain extraction using various publicly available datasets. While commendable, relying on one test dataset limits the study's generalizability. K-fold cross-validation would have been more appropriate for assessing the method's generalizability.

Originality and Significance: Although the proposed method incorporates CRF as RNN to UNet, it relies on previously developed methods, making its novelty limited. More statistical tests are necessary to support the results and validate the study for publication.

Data: The study uses publicly available datasets for the proposed method. However, the authors should provide more details on the datasets used, such as the number of images, their resolution, and the types of abnormalities included, to better understand the study's scope and relevance.

Methodology: While the gradcam based explanation adds value, the proposed method lacks novelty due to previous methodological developments. The computational cost of adding CRF should be discussed, and the feasibility of the proposed method should be established. The study could benefit from a more in-depth discussion of the limitations and challenges of the proposed method, as well as potential ways to address these limitations and extend the method's applicability to other brain extraction scenarios.

Overall Comments:

The study could have improved its generalizability by discussing the proposed method's performance across multiple datasets through K-fold cross-validation, instead of relying solely on one test dataset.

Although the authors cited the BrainMaGe method, they did not utilize it in their study. BrainMaGe is capable of processing healthy brains and including its results would have provided additional insights.

To enhance the readers' understanding, the passage written from line 210-235 should be revised by providing relevant citations and presenting the information in a clearer manner.

Are the changes in the results statistically significant compared to other methods? At least for Hausdorff distance?

This study is limited in that it only applies to healthy brains, while the majority of research work in the field involves brains with abnormalities. Evaluating the proposed method on brains with abnormalities, such as Parkinson's/Alzheimer's, would be beneficial.

Additional Experiments Needed:

The study should include latency analysis to enable readers to understand the time required for running the proposed method. Comparing available methods on CPU would be fair.

Running brainmage on the chosen test dataset and including the results with HD-BET would be useful.

The authors of BrainMaGe have claimed that it works on healthy brains, and determining whether methods trained on tumors could work on healthy brains would be insightful. It should be a matter of running inference on the code.

The authors should conduct additional statistical tests to support the robustness of the proposed method. Readers should be able to understand the statistical differences through p-values.

The study could also benefit from discussing potential ethical considerations related to the use of deep learning techniques in medical imaging, such as bias in training data, transparency, and interpretability.

Finally, the study's contributions and implications for the field should be discussed in more detail. While the proposed method shows promise for brain extraction from healthy brains, its limitations and potential extensions should be explored to contribute to the field's progress. Additionally, the study's potential impact on clinical practice and patient outcomes should be considered.

Reviewers' comments:

Reviewer #1

1. Expand the model name in the first time usage for EVAC and EVAC+

Answer: Thank you for pointing this out. It stands for Efficient V-Net with Additional CRF layer. (Plus is for the additional negative loss function) We have included the corresponding information in the abstract of the article.

2. Authors can explain why they choose V-Net as their base architecture. What are its pros and cons compared to UNET

Answer: Thank you for this great question. V-Net follows the basic multi-scale encoder-decoder like structure of U-Net. However, it has residual connections within each scale of the encoder and decoder, replaces pooling layers with convolutional layers when changing scale and implements Dice Loss for volumetric segmentation instead of binary cross entropy. A disadvantage could be that while the original U-Net was created for 2D images, V-Net is for 3D, thus having higher memory complexity. We have added more details about this in the main document. (line 76).

3. In line 175, author discussing that there are only few methods that use 3D UNET for medical image segmentation (analysis), which is not true. There are plenty of 3D UNETS available for medical image segmentation, classification and synthesis.

Answer: Good point. This was not our intention. This is now clarified. We corrected the section that led to this issue and added relevant citations. (line 177)

4. Adding multi scale inputs should have increased memory load on the model. Authors can discuss the computational efficiency of the models compared too.

Answer: Thank you for this question. We now clarify this point with a new analysis that focuses on studying the memory needs of the model. This new analysis shows that our changes have a maximum of 0.3% increase in the number of weights, so the impact on the memory is very small. We have included this experiment in the revised document. (line 527)

Reviewer #2 (Remarks to the Author):

1. This paper aims to enhance V-net performance by incorporating multiple scale inputs, CRFasRNN, and a loss function to mitigate potential errors introduced by deep learning methods. While prior research such as Cotr[1] and nnunet [2] have shown superior performance for 3D medical image segmentation, the authors have not mentioned or compared their approach with these methods.

Answer: As mentioned in the article, (line 335) we do not compare general models for 3D medical image segmentation because one of our goals is to create a ready to use model for brain extraction. There are multiple factors that affect the performance of a trained model, and since we argue that our model's accuracy is comparable even with limited data, we find it more appropriate to compare to other pre-developed methods that specialize in brain extraction. However, we are aware of these recent developments in the field of medical image segmentation. Thus, we have added a relevant discussion section in the article. (line 400)

2. In terms of novelties such as multi-scale inputs, CRF and a loss function to fine tune segmentation output, these ideas are very common in this field.

Answer: CRFs are common in the field of medical imaging but not when implemented as RNNs. Our model is innovative because as far as we know we are the first to successfully tune and use the CRFasRNN layer for volumetric medical image segmentation. This is significantly different in that it is faster and has more flexibility from using weights instead of hyperparam

eters. Thus, we would like to point out that our idea is not common in this field. Note also in a previous study [3], the authors pointed out that the CRFasRNN approach was not beneficial. We clearly show that this is not the case if we tune the layer to match gray scale volumetric medical image segmentation. (Section 6.2 and 6.3) We hope we can persuade researchers to start looking again at RNN implementations of CRFs for such problems as there is a clear benefit of dynamically updating labels for static segmentation problems.

Reviewer #3

Validity: The authors propose their novel method and apply it to the task of brain extraction using various publicly available datasets. While commendable, relying on one test dataset limits the study's generalizability. K-fold cross-validation would have been more appropriate for assessing the method's generalizability.

Answer: Thank you for this great question. Showing generalizability is a major aspect of this work. However, note that to make the method generalizable we use two datasets for quantitative analysis and four training datasets for qualitative analysis. Thus, while we do agree K-fold cross-validation is a useful technique when we have access to limited test datasets, we believe the test datasets are adequate to show the model's generalizability. Nonetheless, thanks to the review we have acknowledged that we did not point this out in writing but only in figures, so have added relevant information. (line 628)

Originality and Significance: Although the proposed method incorporates CRF as RNN to UNet, it relies on previously developed methods, making its novelty limited. More statistical tests are necessary to support the results and validate the study for publication.

Answer: Thank you for this question that helped us clarify the innovation of our approach. While our method is based on previously developed methods, we would like to point out that previous research on the CRFasRNN layer for medical image segmentation has been giving negative results for 3D medical images saying that the CRFasRNN is not useful. We resolve thi

s misunderstanding. The CRFasRNN layer does improve the results as one would expect. We show this for the first time in this work. As suggested, we have added p-values from Wilcoxon's signed rank test method to emphasize the significance in the difference of metric scores between methods. (Table 2)

Data: The study uses publicly available datasets for the proposed method. However, the authors should provide more details on the datasets used, such as the number of images, their resolution, and the types of abnormalities included, to better understand the study's scope and relevance.

Answer: We have added more information about the datasets, however please note that all datasets used are already published and publicly available. (line 621) [4, 5, 6]

Methodology: While the gradcam based explanation adds value, the proposed method lacks novelty due to previous methodological developments. The computational cost of adding CRF should be discussed, and the feasibility of the proposed method should be established. The study could benefit from a more in-depth discussion of the limitations and challenges of the proposed method, as well as potential ways to address these limitations and extend the method's applicability to other brain extraction scenarios.

Answer: Thank you for these suggestions. Due to implementation limitations (compiler changes), currently it is not feasible to run the model in CPU. However, on GPU it only has a 2.5x fold increase in execution time, suggesting that with the right implementation, the computational cost should not be a major issue. As you suggested, we have added more information about the limitations of the study in the discussion. Novelty is addressed in the "Originality and Significance" section.

Overall Comments:

The study could have improved its generalizability by discussing the proposed method's performance across multiple datasets through K-fold cross-validation, instead of relying solely on one test dataset.

Answer: As mentioned in the validation section, we do not believe K-fold cross-validation is appropriate in this situation.

Although the authors cited the BrainMaGe method, they did not utilize it in their study. BrainMaGe is capable of processing healthy brains and including its results would have provided additional insights.

Answer: As suggested, we have added BrainMaGe in our results. (Figure 5, 6, 7 and Table 2)

To enhance the readers' understanding, the passage written from line 210-235 should be revised by providing relevant citations and presenting the information in a clearer manner.

Answer: We have revised our paper to make our point more clear. Efficient Conditional Random Fields, the technique used in the original paper is an extension of Conditional Random Fields with Mean Field approximation technique. After a thorough review of the literature, we found no other papers that are relevant or helpful that could be cited to help with the explanation. (Section 2.2)

Are the changes in the results statistically significant compared to other methods? At least for Hausdorff distance?

Answer: Addressed in the "Originality and Significance" section. In summary, we show that our method has statistically significant advantages compared to other methods in various metrics across the two datasets.

This study is limited in that it only applies to healthy brains, while the majority of research work in the field involves brains with abnormalities. Evaluating the proposed method on brains with abnormalities, such as Parkinson's/Alzheimer's, would be beneficial.

Answer: Our qualitative analysis on Parkinson's and Alzheimer's in Figure 5 shows that our model even though trained on healthy brains works well on brains with abnormalities. This is something desirable in the neural networks literature. Most models will work well only with th

e types of data that are trained on and will fail on unseen data e.g. unhealthy populations.

Additional Experiments Needed:

The study should include latency analysis to enable readers to understand the time required for running the proposed method. Comparing available methods on CPU would be fair.

Answer: Addressed in Methodology section and above. In summary, users should have no constraints running our method in GPU (approximately 17 seconds per image) but current implementation restrictions due to compiler changes limits CPU usage. Our future investigation is towards simplifying the model while preserving accuracy.

Running brainimage on the chosen test dataset and including the results with HD-BET would be useful. The authors of BrainMaGe have claimed that it works on healthy brains, and determining whether methods trained on tumors could work on healthy brains would be insightful. It should be a matter of running inference on the code.

Answer: Addressed above. In summary, we updated the document with a BrainMaGe comparison.

The authors should conduct additional statistical tests to support the robustness of the proposed method. Readers should be able to understand the statistical differences through p-values.

Answer: Addressed above. We have added p-values to emphasize the significance of the difference between our method and other state of the art methods.

The study could also benefit from discussing potential ethical considerations related to the use of deep learning techniques in medical imaging, such as bias in training data, transparency, and interpretability.

Answer: We appreciate your advice. We have added a paragraph on this topic in the discussion section. (line 420)

Finally, the study's contributions and implications for the field should be discussed in more detail. While the proposed method shows promise for brain extraction from healthy brains, its limitations and potential extensions should be explored to contribute to the field's progress. Additionally, the study's potential impact on clinical practice and patient outcomes should be considered.

Answer: Thank you for these suggestions. We have added more explanation in the discussion section (line 392) to make our model's capability and limitations more clear.

Summary:

We have marked the parts that we have edited or added in the article as red. Table 2. was added to the manuscript as well, along with some citations. We highly appreciate all the reviews.

References

1. Xie, Yutong, et al. "Cotr: Efficiently bridging cnn and transformer for 3d medical image segmentation." *Medical Image Computing and Computer Assisted Intervention–MICCAI 2021: 24th International Conference, Strasbourg, France, September 27–October 1, 2021, Proceedings, Part III 24*. Springer International Publishing, 2021.
2. Isensee, Fabian, et al. "nnu-net: Self-adapting framework for u-net-based medical image segmentation." *arXiv preprint arXiv:1809.10486* (2018).
3. Monteiro, Miguel, Mário AT Figueiredo, and Arlindo L. Oliveira. "Conditional random fields as recurrent neural networks for 3d medical imaging segmentation." *arXiv preprint arXiv:1807.07464* (2018).
4. Glasser, Matthew F., et al. "The minimal preprocessing pipelines for the Human Connectome Project." *Neuroimage* 80 (2013): 105-124.
5. Souza, Roberto, et al. "An open, multi-vendor, multi-field-strength brain MR dataset and analysis of publicly available skull stripping methods agreement." *NeuroImage* 170 (2018): 482-494.

6. Puccio, Benjamin, et al. "The preprocessed connectomes project repository of manually corrected skull-stripped T1-weighted anatomical MRI data." *Gigascience* 5.1 (2016): s13742-016.

REVIEWERS' COMMENTS:

Reviewer #1 (Remarks to the Author):

The authors have carefully considered our feedback and made the necessary modifications to enhance the quality and clarity of your work.

Reviewer #2 (Remarks to the Author):

Thank the authors for their careful revision. The manuscript has now well improved. All the concerns I have raised have been addressed.

Reviewer #2 has also confirmed that the comments of Reviewer #3 appear to have been adequately addressed.